# Tumor Doubling Time Using CT Volumetric Segmentation in Metastatic Adrenocortical Carcinoma

Sarah N. Fuller [1], Ahmad Shafiei [2], David J. Venzon [3], David J. Liewehr [3], Michal Mauda Havanuk [4],
Maran G. Ilanchezhian [1], Maureen Edgerly [5], Victoria L. Anderson [4], Elliot B. Levy [4], Choung D. Hoang [6],
Elizabeth C. Jones [2], Karlyne M. Reilly [1], Brigitte C. Widemann [1], Bradford J. Wood [4], Hadi Bagheri [2]
and Jaydira Del Rivero [1,7,*]

1    Pediatric Oncology Branch, Rare Tumor Initiative, Center for Cancer Research, National Cancer Institute,
     National Institutes of Health, Bethesda, MD 20892, USA; sfuller10722@gmail.com (S.N.F.);
     maran.ilanchezhian@nih.gov (M.G.I.); reillyk@mail.nih.gov (K.M.R.); widemanb@mail.nih.gov (B.C.W.)
2    Radiology and Imaging Sciences, National Institutes of Health Clinical Center, Bethesda, MD 20892, USA;
     ahmad.shafiei@nih.gov (A.S.); ejones@cc.nih.gov (E.C.J.); mohammad.bagheri@nih.gov (H.B.)
3    Biostatistics and Data Management Section, National Cancer Institute, National Institutes of Health,
     Bethesda, MD 20892, USA; venzond@mail.nih.gov (D.J.V.); liewehrd@mail.nih.gov (D.J.L.)
4    Center for Interventional Oncology, National Institutes of Health, Bethesda, MD 20892, USA;
     michal.maudahavakuk@nih.gov (M.M.H.); victoria.anderson@nih.gov (V.L.A.); levyeb@cc.nih.gov (E.B.L.);
     bwood@cc.nih.gov (B.J.W.)
5    Center for Cancer Research, National Cancer Institute, National Institutes of Health,
     Bethesda, MD 20892, USA; edgerlym@mail.nih.gov
6    Thoracic Surgery Branch, National Cancer Institute, National Institutes of Health, Bethesda, MD 20892, USA;
     chuong.hoang@nih.gov
7    Developmental Therapeutics Branch, Center for Cancer Research, National Cancer Institute, National
     Institutes of Health, Bethesda, MD 20892, USA
*    Correspondence: jaydira.delrivero@nih.gov

**Abstract:** Adrenocortical carcinoma (ACC) is a rare malignancy with an overall unfavorable progno-
sis. Clinicians treating patients with ACC have noted accelerated growth in metastatic liver lesions
that requires rapid intervention compared to other metastatic locations. This study measured and
compared the growth rates of metastatic ACC lesions in the lungs, liver, and lymph nodes using
volumetric segmentation. A total of 12 patients with metastatic ACC (six male; six female) were
selected based on their medical history. Computer tomography (CT) exams were retrospectively
reviewed and a sampling of ≤5 metastatic lesions per organ were selected for evaluation. Lesions
in the liver, lung, and lymph nodes were measured and evaluated by volumetric segmentation.
Statistical analyses were performed to compare the volumetric growth rates of the lesions in each
organ system. In this cohort, 5/12 had liver lesions, 7/12 had lung lesions, and 5/12 had lymph
node lesions. A total of 92 lesions were evaluated and segmented for lesion volumetry. The volume
doubling time per organ system was 27 days in the liver, 90 days in the lungs, and 95 days in the
lymph nodes. In this series of 12 patients with metastatic ACC, liver lesions showed a faster growth
rate than lung or lymph node lesions.

**Keywords:** adrenocortical carcinoma; doubling time; growth rate; neoplasm metastasis;
volumetric segmentation

## 1. Introduction

Adrenocortical carcinoma (ACC) is a rare endocrine malignancy with an estimated
incidence of 1.5–2 per million people per year. It has a very poor prognosis with an overall
5-year mortality rate of 75–90% and an average survival from the time of diagnosis of
14.5 months [1,2]. Complete open resection of the primary tumor with negative surgical
margins confers the best prognosis, however local recurrence and/or metastatic disease
is still common even in this scenario [3]. Metastatic disease on presentation is seen in

25–30% of patients and is a key prognostic factor associated with poor outcomes. ACC most commonly metastasizes to the lung (40–80%), liver (40–90%), and bone (5–20%), and resection of metastatic lesions can improve 5-year survival, reduce pain, and control hormone production [2–6]. Single institution studies have shown that patients who undergo local treatment of liver metastases have a 5-year survival of 39–51% while patients who undergo resection of lung metastasis have a 5-year survival of 41%, with particular benefit in younger patients [7–10]. Prospective and retrospective studies indicate that mitotane monotherapy or combined therapy of mitotane with cytotoxic drugs such as etoposide, doxorubicin, and cisplatin may increase progression-free and recurrence-free survival in patients with adrenocortical cancer [11,12].

In many cancers, the growth rate of metastases is used as a prognostic factor that guides treatment choice [13–15]. Serial analyses of imaging studies can be used to determine tumor doubling time. However, two-dimensional tumor measurement of the short and long axis is susceptible to intra- and inter-observer variation and has limited utility in the assessment of irregularly shaped tumors [16]. An alternative method of tumor evaluation is volumetric measurement, which can generate more accurate shape calculations in lesions [17]. Although volumetric segmentation has been considered the "gold standard" of cancer imaging by some authors, it is time- and resource-intensive, which makes it impractical for routine daily usage in clinical practice [18–21]. However, this is a more accurate and reliable method in the study of rare cancers such as ACC where the patient population is limited.

This study aimed to quantify the growth pattern and doubling time of metastatic ACC lesions to the liver, lung and lymph nodes using volumetric analysis of serial computer tomography (CT) images. Primary adrenal carcinomas were not evaluated in this study; rather, metastatic lesions in the major organ hosts in liver, lung, and lymph nodes were compared to investigate differences in growth rates across organ systems.

## 2. Materials and Methods

### 2.1. Patient Selection

A retrospective chart review at the National Institutes of Health identified 56 patients with a pathologic confirmation of primary ACC from 1999 to 2019. Consent was obtained to participate in an institutional review board (IRB)-approved investigational study at time of treatment. Additionally, the present analysis was also performed under another IRB-approved retrospective reporting protocol that met the criteria for waiver of further consent.

An independent review of medical records by two investigators identified 56 patients with metastatic ACC. Forty patients who were on active systemic treatment were excluded from this study to exclude variations in treatment response. In total, 16 out of 56 patients were identified as not receiving systemic therapy during the evaluation window. A total of 5 of the 16 patients with metastatic lesions had a history of systemic therapy > 1 year preceding the evaluation window. Subject 1 had treatment consisting of PSC Velban, mitotane 3 years prior; subject 2 had records that indicated MAVE treatment from June 1999–Feb 2000, subject 4 had treatment consisting of tariquidar/MAVE (mitotane, Adriamycin, vincristine, and etoposide) × 6 cycles 2 years prior to evaluation window and Gemcitabine and cisplatin × 6 cycles 1 year prior. Subject 10 was initially treated with cyclophosphamide, vincristine, and dacarbazine (CVD) × 6 cycles for presumed metastatic pheochromocytoma and then pembrolizumab (based on her history of Lynch syndrome and loss of expression for MSH-2 and MSH-6 in her tumor cells). She then presented to our institution and the diagnosis of ACC was confirmed and she received EDP and mitotane. However, this treatment was started after the measurement period [22].

Four patients who were included in our study (cases 1, 3, 4 and 7) received locoregional therapy during the interval of study; however, treated lesions were excluded from our growth rate evaluations. Eight patients were on mitotane. In all patients, mitotane levels were <6 μg/mL and since there were sub-therapeutic levels it is unclear, however, not likely, to have had an effect on the growth rate.

Eligible subjects had metastatic ACC with two CT scans of chest/abdomen/pelvis performed within a 1–6-month interval. These subjects were not receiving active systemic therapy during the surveillance window. Of the 16 patients identified with the appropriate medical criteria, 3 were excluded due to unavailable serial CT scans within the specified evaluation time frame of 1–6 months. The remaining 13 patients had appropriate CT imaging, and 3 patients within this group had additional [18]FDG-PET scans. [18]FDG-PET is highly sensitive for detecting metastatic disease in ACC, yet only 3 in our cohort had the aforementioned scan because it was not widely available >15 years ago for patients undergoing treatment. Moreover, this imaging modality is underutilized in ACC patients since it is not routinely reimbursed by health insurance [23–25]. After reviewing the PET scans from these three patients, one patient was included due to [18]FDG-PET confirmation of ACC metastatic lesions and another patient had avid pulmonary lesions but non-avid liver lesions, therefore only the pulmonary lesions were included in the analysis. The third patient was excluded due to negative PET lesions. Ultimately, 12 patients (6 males; 6 females) were included, with an average age of diagnosis at 50 years of age (range, 31–70 years) (Figure 1). Eight patients were alive at the time of the analysis.

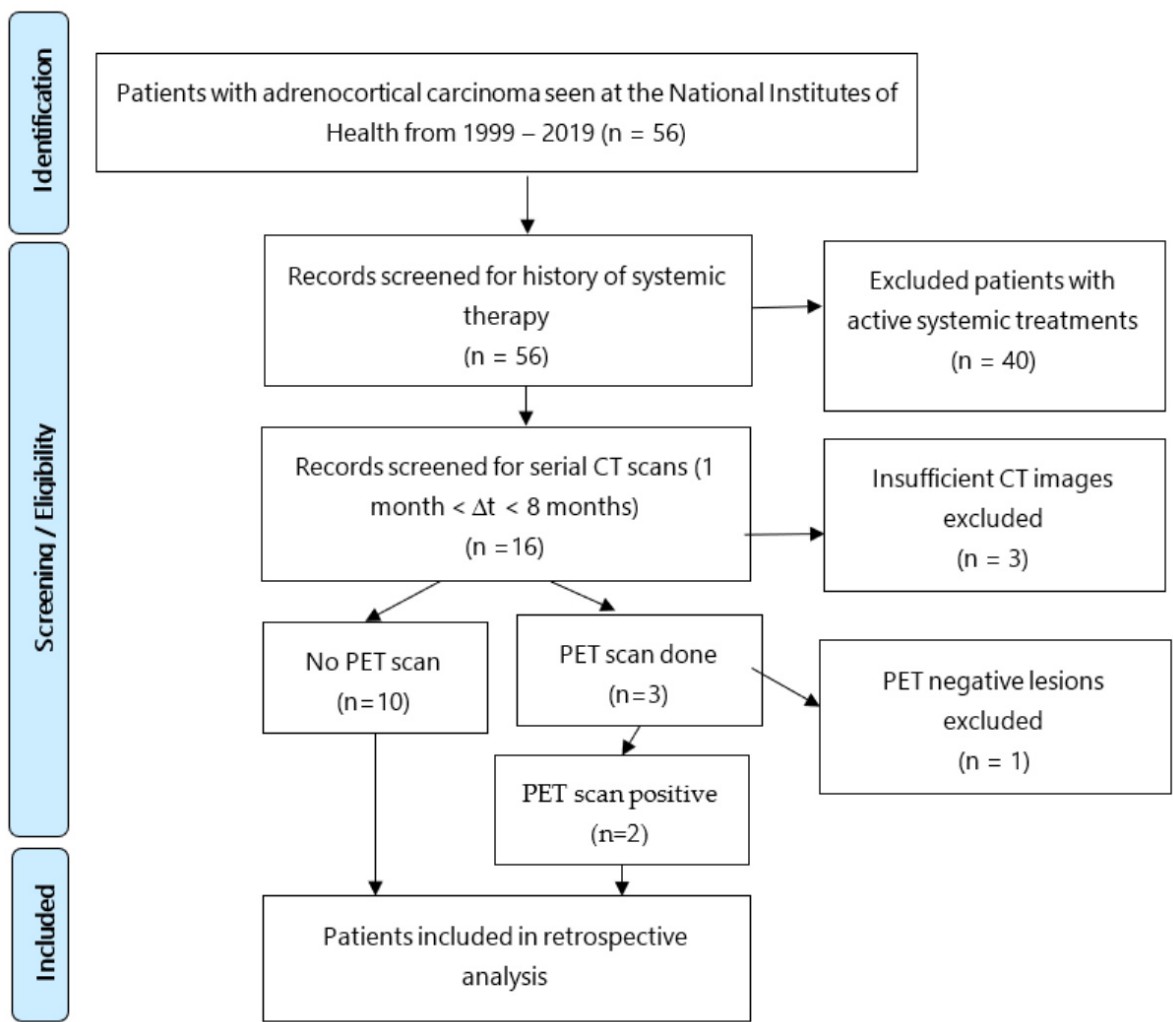

**Figure 1.** Patient selection process for retrospective growth rate/doubling time analyses of patients with metastatic adrenocortical cancer (ACC).

### 2.2. Imaging Analysis

Axial soft tissue window series of chest/abdomen/pelvis CT scans with slice thickness of 2–10 mm was selected for evaluation. Portal venous phase CT series were used when available. A non-contrast axial CT series was used when the lesions were clearly evaluable, for example in the lung, and the radiologist was confident about the lesion's boundary for segmentation. All lung lesions were evaluated with the standard lung window and kernel setting, and the rest of the lesions were measured and segmented in soft tissue series generated with a soft tissue kernel.

All baseline and follow-up lesions were measured two-dimensionally and segmented using the Carestream picture archiving and communication system (PACS) 11.1 software (Rochester, NY, USA). Lesion inclusion criteria were as follows: lesions in the liver and soft tissue with a long axis diameter of ≥10 mm, lymph node lesions having a short axis diameter of ≥10 mm, and pulmonary lesions having a long axis of ≥6 mm. Up to five lesions per organ system were recorded to provide enough sampling of the metastatic lesion growth rate (Figures 2 and 3).

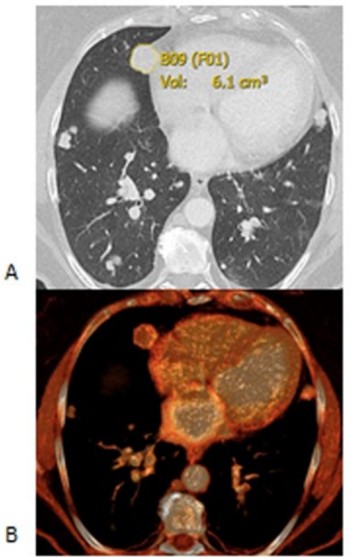
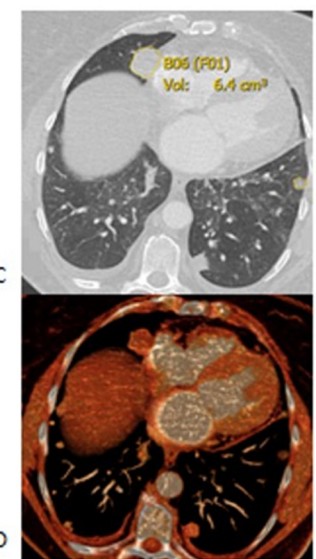

**Figure 2.** Axial CT scans at the baseline (**A**,**B**) and follow up (**C**,**D**) after 35 days. (**A**,**B**) are axial lung window images. The segmented right lung nodule was 6.1 cm3 in the first scan (**A**) and it grew to 6.4 cm$^3$ in the follow up scan (**B**). (**C**,**D**) are the corresponding multiplanar volume rendering images.

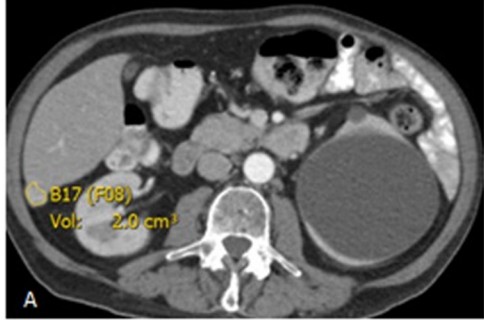
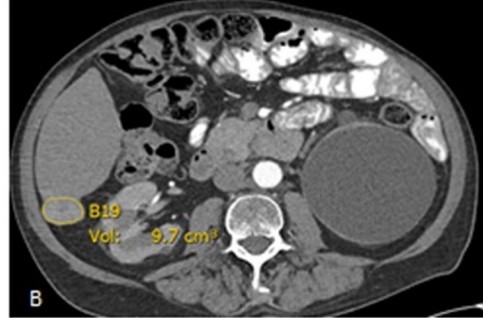

**Figure 3.** Axial post-contrast CT scan demonstrates an enhancing lesion in the posterior right hepatic lobe, growing from 2.0 cm$^3$ at the baseline (**A**) to 9.7 cm$^3$ in the follow up (**B**) after 102 days.

Volumetric segmentation was performed manually by tracing the lesion within each slice and was carried in Carestream PACS using the Livewire Segmentation Mode tool by a postdoc with 3 years' experience in image processing (A.S.) and was supervised by a radiologist with 27 years of experience (H.B.). After finalizing the segmentation

on each slice, the lesion volume was automatically calculated and saved in the PACS. Lesions were classified by the organ involved into 4 categories: Liver (L), Pulmonary (P), Lymph Node (N), and other locations (O).

### 2.3. Statistical Analysis

Because of the instability of doubling times in potentially slow-growing lesions, the growth rates (GR) were analyzed with an exponential (base 10) lesion growth model. A mixed model analysis of variance was used to analyze growth rates. The lesion location (liver, lymph node or lung) was specified as a fixed effect, and patient by location was specified as the random effect. Deviations from expected values were graphically examined for normality and homoscedasticity; residual variance was partitioned because of heteroscedasticity (liver vs. lymph node and lung). The Satterthwaite approximation for the denominator degrees of freedom was utilized. Lesion growth rates and their paired differences were estimated with model-based LS-Means, and uncertainty was estimated with 95% confidence intervals (CIs for differences are Bonferroni adjusted). For the lesion effect only model, the null hypothesis of equal rates was assessed by an *F*-test. Additional *F*-tests were performed using age at diagnosis and sex as single factors included in the location model. Age was modeled as a covariate having a linear trend effect. Equal effects across locations were assumed. Additional analyses are included in the Supplemental Statistical Section Tables S1 and S2.

### 3. Results

The final patient cohort included 12 patients, 6 males and 6 females: 5/12 had a total of 11 liver lesions, 7/12 had 27 lung lesions, 5/12 had 6 lymph node lesions, and 2/12 patients had other retroperitoneal recurrence (patients 6 and 7); 5/12 patients had two lesion locations (Table 1). We used a total of 24 CT examinations for baseline and follow up evaluation volumetric segmentation of 88 lesions. In 18/24 of the scans, portal venous phase CT series were used for measurement and volumetry of 72 lesions. Non-contrast CT was used in 6/24 scans for measurement and volumetry of 24 lesions. Slice thickness was constant at the baseline and follow up scans in 11/12 patients. In one patient the baseline scan was in 2.5 mm thickness and 2 mm in the follow up scan. Slice thickness was 10 mm in two patients for the baseline and follow up scans (4/24 scans) due to older scanning techniques dated in 1998 and 1999. Image thickness was 5 mm in six patients, and 2 mm in three patients. Metastatic lesions found in locations other than the lung, liver, or lymph nodes were excluded from analysis due to small sample size. This resulted in the exclusion of four retroperitoneal lesions across two patients. The outcome analyzed was the growth rate of the exponential (base 10) volume growth model (Figure S4).

The volume doubling time per organ system was 27 days in the liver, 90 days in the lungs, and 95 days in lymph nodes. Liver lesions exhibited the fastest growth rate compared to other locations (0.34 vs. 0.095 vs. 0.10; *F*-test *p* = 0.025). Liver lesions had a CI of 0.177–0.504 (Table 2 and Figures S1 and S2). As an indication of the scale of the growth rates, note that an increase of 0.1 in the lymph node (N) or pulmonary (P) base 10 exponential growth rate would divide the corresponding volume doubling time by two. We compared the liver lesion growth rates (L) and the lymph node lesion growth rates (N) and found liver lesions grew significantly faster with the 95% CIs excluding the null hypothesis of zero. A similar result was found when comparing liver lesion growth rates (L) to pulmonary lesion growth rates (P) (Table 2). Confidence intervals were calculated (Figures S3 and S4). Of note, there were two instances of similar growth rates in the liver and the other two organs (Figures S1 and S2). The two lesions in the "other" category, outside of the liver, lymph nodes, and pulmonary systems, were excluded due to limited numbers.

**Table 1.** Patient demographic characteristics. MAVE = mitotane, Adriamycin, vincristine, and etoposide.

| ID | Sex and Age at Diagnosis | Metastatic Lesions Per Organ | | | | ΔT * (Days) | Mitotane | Mitotane Levels (μg/mL) | Systemic Therapy Treatment History |
|----|----|----|----|----|----|----|----|----|----|
| | | Liver | Lung | Lymph Nodes | Other | | | | |
| 1 | M33 | 0 | 4 | 2 | 0 | 63 | Yes | 3.3 | PSC VALS 5CY, mitotane Velban × 5 cycles, |
| 2 | M31 | 0 | 5 | 0 | 0 | 55 | Yes | 3.7 | Records unclear, possibly on MAVE from June 1999–Feb 2000 |
| 4 | F46 | 0 | 4 | 1 | 0 | 56 | Yes | 4.1 | Tariquidar/MAVE (mitotane, Adriamycin, vincristine, and etoposide) × 6 cyclesGemcitabine and cisplatin × 6 cycles. |
| 5 | M32 | 5 | 0 | 0 | 0 | 36 | Yes | 1.1 | None prior |
| 6 | F58 | 0 | 2 | 0 | 0 | 175 | No | - | None prior |
| 7 | F36 | 1 | 0 | 1 | 1 | 60 | Yes | 1.1 | None prior |
| 8 | M55 | 0 | 3 | 0 | 1 | 171 | Yes | 5.9 | None prior |
| 9 | M67 | 2 | 0 | 0 | 0 | 100 | Yes | 5.0 | Mitotane likely held with subtherapeutic levels, medical records unclear |
| 10 | F57 | 0 | 4 | 0 | 0 | 91 | No | - | None during measurement period |
| 11 | F57 | 0 | 0 | 1 | 0 | 191 | Yes | <1.0 | None prior |
| 12 | F70 | 2 | 5 | 0 | 0 | 33 | No | - | None prior |
| 14 | M61 | 1 | 0 | 1 | 0 | 90 | No | - | None prior |

ΔT * = change in time.

**Table 2.** Estimated (base 10) growth rates by lesion location and differences of lymph node and lung rates from rates of the liver. CIs for differences are Bonferroni adjusted.

| Lesion | Estimate | 95% CI | Diff from L | 95% CI |
|----|----|----|----|----|
| L | 0.341 | (0.177, 0.504) | - | - |
| N | 0.095 | (0.058, 0.132) | 0.246 | (0.034, 0.458) |
| P | 0.101 | (0.071, 0.130) | 0.240 | (0.029, 0.451) |

As secondary aims, we tested degrees of associations between growth rate as compared to sex and age at diagnosis, while adjusting for location (Supplemental Statistical Section Table S2). Weak associations were found for sex ($p = 0.67$, *F*-test) and age ($p = 0.39$, *F*-test), indicating a faster growth in females and the elderly.

## 4. Discussion

This retrospective analysis of ACC assessed the growth rate of metastatic lesions to the liver, lung, and lymph nodes in patients with stabilization of tumor growth and we followed the interval growth until disease progression that requires further treatment. Lesion doubling time in the liver exhibited a faster growth rate than either the lungs or the lymph nodes. Only weak associations were detected with higher age at diagnosis and female sex. As the care of each individual patient can vary depending on case specifics and the location of their metastatic burden, the data collected from this cohort may alert clinicians to the importance of actively monitoring metastatic ACC liver lesions particularly closely for signs of rapid growth compared to other organ systems. Extensive research into ACC metastatic lesion behavior is limited given the rarity of this neoplasm, however others have emphasized the severity of liver metastases when present. Gaujoux et al. reported that surgical resection of ACC liver metastases was an independent positive

predictive factor for survival and reported a median disease-free and overall survival after hepatectomy of 7 and 31.5 months, respectively, with a 5-year survival of 39% [8]. Ripley et al. also reported survival benefits in patients with metastatic ACC who underwent resection or radiofrequency ablation of liver lesions with 5-year actuarial survivals of 29% and 29%, respectively [26]. Furthermore, Mauda-Havakuk et al. reported that image-guided locoregional imaging including the liver is associated with prolonged survival [27]. These and other studies argue for an aggressive approach to liver metastases as to improve survival within the context of worse prognosis with advanced liver disease [28,29].

Prior reports offer valuable insight into tumor growth rate patterns, yet we found no study that evaluated the differences in growth as a function of the organ system. Tanaka et al. reported that patients with a hepatic colorectal cancer metastasis doubling time of >45 days had few liver recurrences when compared with those having doubling time of <45 days; the latter patients had a higher risk for multiple early recurrences and a prediction of poor prognosis [30]. A meta-analysis study by Nathani et al. reported an HCC doubling time of approximately 4–5 months, with heterogeneity depending on risk factors identified [31]. Miura et al. similarly used a formula to calculate the volume doubling times of solid and whole tumor components in patients with non-small cell lung cancer and concluded that a solid tumor doubling time of <400 days is a poor prognosis indicator when compared with tumor doubling times of >400 days [32]. Our study uniquely offers a comparison of metastatic ACC lesions behavior between organ systems despite a limited cohort.

Our methodology is in line with prior studies, as we aimed to characterize tumor doubling times in selected patients off-treatment, using imaging studies at two time points and setting a prior time restriction between CT comparisons [33–37]. Mehrara et al. highlighted the difficulties of using doubling time for tumor growth, particularly in the setting of short measurement time intervals and uncertainties in tumor volume. To overcome any possible inaccuracies of calculating the lesion volume by the Schwarz's formula, we aimed to reduce this limitation by using volumetric segmentation analysis.

ACC lesions can demonstrate aggressive growth and the interval of 1–12 months between CT scans was considered representative. According to RECIST 1.1 guidelines, follow-up at 6–8 weeks interval is appropriate when the beneficial effect of therapy is not known during a Phase II trial. For the purposes of this retrospective study, a wide time interval is justified given the known heterogeneity in growth patterns across patients [38].

We chose to target our investigation to a population with no active systemic therapy to determine the growth during the time when systemic treatment was not indicated, in order to characterize baseline metastatic lesion growth without intervention. Excluding the patients on treatment regimens greatly reduced our sample size. In future studies, it would be informative to compare growth rates of metastatic lesions under various systemic therapy options. This information would aid in further understanding the effect of treatment and identify whether the location of the metastatic disease plays a role in the lesion's response to treatment. Future studies should also include patients with lesions outside of the liver, lung, and lymph nodes to reliably estimate growth rates in soft tissue locations.

*Limitations*

This study is limited by patient sample size due to the rare nature of ACC, as well as the non-random selection of patients. Moreover, this limited number of patients may have underestimated the accurate growth rate of the metastatic lesions, since only those with slower growing lesions are recommended to undergo careful monitoring of lesions with serial imaging studies instead of rapid intervention. ACC is a heterogenous disease and varies from an aggressive course with patient survival measured in months to a more indolent course with patients living with the disease for years. The basis for these differing clinical presentations is not known and tumor classification methods with better clinical prognostic value are needed to help rationally guide the clinical management of patients

with ACC. Our study excluded all patients currently undergoing therapy, which further restricted our selection to patients that had indolent tumors at the baseline. Treatment typically involves surgical resection of the primary tumor, and in advanced disease not amenable to radical resection, cytotoxic drugs can be added to mitotane [5]. In our cohort, four patients received systemic therapy prior to our window of CT scan review, which superimposes another theoretical complication of patients continuing to experience the benefits of chemotherapy even months after treatment had been discontinued.

Lesion selection bias may also have been a factor, since lesions requiring interventional radiology treatment during the chemotherapy-free window were necessarily excluded from our evaluation. Four patients fell in this category of receiving radiofrequency ablation or targeted therapy during the evaluation window. Excluding the more concerning lesions likely predisposed our data to reflect more conservative growth rates than would be seen in a general patient stream and to diminish the true differential of growth rates. Furthermore, although ablated lesions were excluded from our analysis, there is an unlikely possibility that the intervention affected the tumors of interest for this study in a spill-over effect on neighboring target lesions. [18]FDG-PET scans were available for only three patients, which restricted our ability to confirm ACC metastatic lesions using this imaging modality. Another imaging limitation of this study is that in 6 out of 24 CT scans, we had to use non-contrast CT series, as these were the only available imaging studies on those specific time points. We included these scans only if the radiologist of the team was confident about the lesion boundaries for measurement, segmentation, and volumetry. One additional imaging limitation was that we had to use thick slice CT scans in two patients (and four scans) as those scans were old and thin section images were not available.

## 5. Conclusions

This retrospective analysis used volumetric segmentation in CT examinations to assess the growth rates of ACC metastatic lesions to the liver, lung, and lymph nodes in 12 patients where systemic treatment was not indicated. Lesions in the liver grew the fastest, with an expected doubling time of 27 days as compared to the lungs or liver, which had doubling times of 90 and 95 days, respectively. Based on this limited observation, metastatic ACC liver lesions should be monitored closely for signs of rapid growth to guide us on treatment approaches.

**Supplementary Materials:** The following are available online at https://www.mdpi.com/article/10.3390/curroncol28060370/s1, Figure S1: Scatter plot of exponential growth rate vs. case number (patient ID, Table 1) by type of lesion (L = Liver, N = Lymph Node, P = Pulmonary (Lungs)), Figure S2: Box-and-whisker plots of exponential growth rates by lesion location, Figure S3: Mixed model LS-Means (reference lines) and their 95% Cis, Figure S4: Differences between LS-Means and their 95% CIs (Bonferroni adjusted), Table S1: Case by Location, Table S2: Mixed model ANOVA results.

**Author Contributions:** Conceptualization, S.N.F., H.B. and J.D.R.; methodology S.N.F., D.J.V., H.B. and J.D.R.; formal analysis D.J.V. and D.J.L.; investigation S.N.F., A.S., H.B. and J.D.R.; data curation, S.N.F., D.J.V., D.J.L., H.B. and J.D.R.; writing—original draft preparation S.N.F. and J.D.R.; writing—review and editing D.J.V., M.M.H., M.G.I., M.E., V.L.A., E.B.L., C.D.H., E.C.J., K.M.R., B.C.W., B.J.W., H.B. and J.D.R.; supervision, H.B. and J.D.R. All authors have read and agreed to the published version of the manuscript.

**Funding:** This project was funded in whole or in part with federal funds from MyPART: My Pediatric and Adult Rare Tumor Network-Cures. ZIA BC 011852. The content of this publication does not necessarily reflect the views or policies of the Department of Health and Human Services, nor does mention of trade names, commercial products, or organizations imply endorsement by the US Government. Funded by the NCI Intramural Research Program.

**Institutional Review Board Statement:** The study was approved by the Institutional Review Board of NIH IRB (protocol code: 19-C-0016 and 20-C-N095 and date of approval: 21 January 2019 and 21 September 2020).

**Informed Consent Statement:** IRB-approved retrospective reporting protocol that met criteria for waiver of further consent.

**Data Availability Statement:** The data presented in this study are available in this article and the Supplementary Materials.

**Conflicts of Interest:** The authors declare no conflict of interest.

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
