# Peer review of "Tumor Doubling Time Using CT Volumetric Segmentation in Metastatic Adrenocortical Carcinoma"

_curroncol, doi:10.3390/curroncol28060370_

Round 1

Reviewer 1 Report

POINTS OF STRENGTH

“This information would aid in further understanding the effect of treatment and identify whether location of the metastatic disease plays a role in the lesion’s response to treatment.”

On the other hand, this paper has some questionable points.

Page 2 line 63

Page 2 line 79

ACC lesions in patients off systemic treatment in liver, lung and lymph nodes

→Some patients are receiving systemic therapy in this study, so I think this sentence should be rewritten.

Page 2 line 82

Who had a history of systemic therapy? I think it is desirable to describe it.

Does a history of systemic therapy appear to affect the rate of tumor growth?

Page 3 line 107-113

Axial soft tissue window series of chest/abdomen/pelvis CT scans with slice thickness of 2–10 mm was selected for evaluation.

All lung lesions were evaluated with standard lung window and kernel setting, and the rest of the lesions were measured and segmented in soft tissue series generated with soft tissue kernel.

→ Axial soft tissue window and standard lung window series were selected for evaluation.

→ It is doubtful how the tumor volume was calculated from the image with a slice thickness of 10 mm.

Page 3 line 116

Lesion inclusion criteria were as follows: lesions in the liver and soft tissue with a long axis diameter of ≥ 10 mm, lymph node lesions having a short axis diameter of ≥ 10 mm, and pulmonary lesions having a long axis of ≥ 6 mm.

→ I think that lesions with increased FDG accumulation on PET are metastatic lesions. However, ten patients have not been imaged with PET. I am concerned that the selection criteria for lesions do not include benign lesions. Did all the selected lesions grow?

.Figure 1: please enter the number of people in all the squares in Figure 1.

Figure 2: A and B are axial lung window images. C and D are multiplanar volume rendering images

→ A and C are axial lung window images. B and D are multiplanar volume rendering images

Please describe in more detail. For example, patient information, lesion location, degree of size change, etc.

Figure 3: Did you create volume rendering images? Figure 2 shows volume rendering images. How was volume rendering images used in this study?

Author Response

We thank very much the reviewer for making several valid, and very important points. We agree with the reviewer comments/suggestions. Please see answers below:

Page 2 line 63

Page 2 line 79

ACC lesions in patients off systemic treatment in liver, lung and lymph nodes

→Some patients are receiving systemic therapy in this study, so I think this sentence should be rewritten.

Response: Thank you for this comment, this line was adjusted to read, “This study aimed to quantify the growth pattern and doubling time of metastatic ACC lesions to the liver, lung and lymph nodes using volumetric analysis of serial computer tomography (CT) images."

---

Page 2 line 82

Who had a history of systemic therapy? I think it is desirable to describe it.

Does a history of systemic therapy appear to affect the rate of tumor growth?

Response: Thank you, we have added descriptions of the systemic therapy that the patients were on within the patient selection discussion, see lines 95-99

---

Page 3 line 107-113

Axial soft tissue window series of chest/abdomen/pelvis CT scans with slice thickness of 2–10 mm was selected for evaluation.

All lung lesions were evaluated with standard lung window and kernel setting, and the rest of the lesions were measured and segmented in soft tissue series generated with soft tissue kernel.

→ Axial soft tissue window and standard lung window series were selected for evaluation.

→ It is doubtful how the tumor volume was calculated from the image with a slice thickness of 10 mm.

 Response: Thank you for this comment. We mentioned in the section of patients and methods, some studies goes back to a time that only 10 mm sections were available. We are aware this is a limitation and we addressed this limitation in the last two sentences in the limitation section. We hope the reviewer find this response adequate. 

---

Page 3 line 116

Lesion inclusion criteria were as follows: lesions in the liver and soft tissue with a long axis diameter of ≥ 10 mm, lymph node lesions having a short axis diameter of ≥ 10 mm, and pulmonary lesions having a long axis of ≥ 6 mm.

→ I think that lesions with increased FDG accumulation on PET are metastatic lesions. However, ten patients have not been imaged with PET. I am concerned that the selection criteria for lesions do not include benign lesions. Did all the selected lesions grow?

Response: Thank you for making a great point. Yes, The lesions in patients who didn’t have a PET were only included if there was a growth over time in constitutive CT scans.

---

Figure 1: please enter the number of people in all the squares in Figure 1.

Response: Thank you, this change has been added.

---

Figure 2: A and B are axial lung window images. C and D are multiplanar volume rendering images

→ A and C are axial lung window images. B and D are multiplanar volume rendering images

Please describe in more detail. For example, patient information, lesion location, degree of size change, etc.

Response: Thanks for you for this suggestion. Additional description of the lesion was added to the figure legend.

--

Figure 3: Did you create volume rendering images? Figure 2 shows volume rendering images. How was volume rendering images used in this study?

Response: Yes, we made the volume-rendering images. We did not use volume-rendering images for our analysis. These images were just added to the figures to provide additional visual impression of the lesion changes during the two time points.

Reviewer 2 Report

The reported manuscript describes a retrospective analysis in metastatic adrenocortical carcinoma (ACC) patients evaluating growth rates of metastatic liver, bone, and lymph-node lesions as assessed reviewing CT scans with the volumetric segmentation technique.  ACC is a rare and aggressive cancer, with poor, but potentially heterogeneous, prognosis. A better characterization of the aggressivity related to different metastatic patterns might help to guide the treatment choice in ACC patients; therefore, the reported study is of potential interest. The study however suffers of several limitations:

A major limitation of the studies is the variable interval of follow-up available in different patients with several patients (often without liver lesions) had follow-up longer than 2-3 months that would be appropriate in patients with metastatic ACC. According with this reviewer, this might greatly affect the different growth rate reported for the liver versus lung and lymph nodes, thus influencing the conclusion.

Patients with recent story of treatment cannot be considered “stable” without treatment and should be excluded from the study.

The mitotane can last in the blood for long time, thus the effect of this drug might persist for long time after drug discontinuation. To exclude the interference of mitotane of growth rate of different lesion authors should exclude the persistence of considerable mitotane levels in the blood (which are generally monitored in these patients) of those patients previously treated with this drug.

I would suggest to the author to explore whether the difference observed in the growth rate of the lesions using volumetric segmentation technique could be observed also evaluating the longer diameter technique of the same lesions.   

Additional minor comments:

Line 70 and 76: All the evaluated primary ACC were metastatic? They did not have any localized tumor? This sound strange and should be clarified

FDG-PET is available only in three patients. It should not be considered.

In figure 1 authors reported PET scan done in 2 patients whereas in the text they wrote 3 patients. Please check correspondence between text and figure.

line 220: please specify the achronim

Author Response

We thank very much the reviewer for making several valid, and very important points. We agree with the reviewer comments/suggestions

The reported manuscript describes a retrospective analysis in metastatic adrenocortical carcinoma (ACC) patients evaluating growth rates of metastatic liver, bone, and lymph-node lesions as assessed reviewing CT scans with the volumetric segmentation technique.  ACC is a rare and aggressive cancer, with poor, but potentially heterogeneous, prognosis. A better characterization of the aggressivity related to different metastatic patterns might help to guide the treatment choice in ACC patients; therefore, the reported study is of potential interest. The study however suffers of several limitations:

A major limitation of the studies is the variable interval of follow-up available in different patients with several patients (often without liver lesions) had follow-up longer than 2-3 months that would be appropriate in patients with metastatic ACC. According with this reviewer, this might greatly affect the different growth rate reported for the liver versus lung and lymph nodes, thus influencing the conclusion.

Response: Thank you for making a very important point. We realized there was a typo on the follow up interval. We corrected the follow up interval to be 1-6 months for eligible subjects. Most patients had a follow up scans interval of 2-3 months and 3 patients had interval follow up of 5-6 months. 

---

Patients with recent story of treatment cannot be considered “stable” without treatment and should be excluded from the study.

Response: Thank you for an important point. We have revised the wording of “stable” throughout the manuscript. The point is well taken and we aimed to be more clear when discussing the patients that have not progressed in their disease even while off treatment for >1 year.

---

The mitotane can last in the blood for long time, thus the effect of this drug might persist for long time after drug discontinuation. To exclude the interference of mitotane of growth rate of different lesion authors should exclude the persistence of considerable mitotane levels in the blood (which are generally monitored in these patients) of those patients previously treated with this drug.

Response: This is a valid point brought up by the reviewer. Mitotane levels were added on the table to exclude interference of mitotane on growth rate. None of the patients had therapeutic levels or were not on mitotane at the time of the analysis 

---

I would suggest to the author to explore whether the difference observed in the growth rate of the lesions using volumetric segmentation technique could be observed also evaluating the longer diameter technique of the same lesions

Response: Thank you do much for this comment. We would like to draw the reviewer's attention that comparing long axis diameter with volume segmentation was not the objective of this work. This comparison has been done before and there are several published articles in the literature on this and we have already referenced them in the (page 2 line 75). So, although volume measurement was a time consuming and tedious process, we considered this over long axis diameter to have a more reliable and accurate method for evaluation of the lesions. Another point to be considered is the fact that the long axis cannot clearly reflect the volume and calculation of doubling time needed a more accurate method. We truly hope this answer is acceptable to the reviewer.

---

Line 70 and 76: All the evaluated primary ACC were metastatic? They did not have any localized tumor? This sound strange and should be clarified

Response: Thank you for pointing this out, this language was adjusted. Moreover, our purpose was to evaluate the growth rate of metastatic lesions and we dit no evaluate local recurrence.

---

FDG-PET is available only in three patients. It should not be considered.

Response: Thank you for pointing this out. Because of rarity of the disease we decided to be more inclusive. Moreover, we believe FDG-PET scan is underutilized as complementation to anatomical imaging studies for diagnosis and staging. We hope the reviewer accept our response not to exclude patients who has a FDG-PET scan and further clarification is in the text. 

---

In figure 1 authors reported PET scan done in 2 patients whereas in the text they wrote 3 patients. Please check correspondence between text and figure.

Response: Thank you for this comment. We certainly agree with the reviewer that our text was a bit confusing because of the 3 patients who had FDG-PET scan, only 2 patients had FDG-PET positive lesions. We wanted to include only the patients with FDG-PET positive lesions. We revised the manuscript to make it easier and clear to read and we hope the reviewer agree with our comments. 

Round 2

Reviewer 1 Report

The manuscript has been well revised. I think this manuscript deserves publication.

Reviewer 2 Report

I do not have additional comments